# Comprehensive Benefit of Crop Straw Return Volume under Sustainable Development Management Concept in Heilongjiang, China

Jia Mao [1], Ziang Zhao [1], Xiangyu Li [2], Honggang Zhao [3,*] and Ciyun Lin [1]

1   School of Transportation, Jilin University, Changchun 130012, China
2   College of Automotive Engineering, Jilin University, Changchun 130012, China
3   College of Chemical Engineering, Xinjiang Normal University, Urumqi 830054, China
*   Correspondence: 107621997010002@xjnu.edu.cn

**Abstract:** Straw burning can cause serious environmental pollution, whereas returning straw to the fields, as a green production method, can improve the rural environment and strengthen the sustainable development of agriculture. According to statistics, China produced 797 million tons of straw in 2020, but the current straw return technology still needs to be improved; the ability of farmers to choose the correct amount of straw to be returned to the field and their awareness of environmental protection still need to be strengthened. Straw is still openly burned in some areas, causing environmental pollution and the waste of resources, which are contrary to the concept of sustainable development in China. In this study, we estimated the amount of straw resources in Heilongjiang Province, a major grain-producing province in China, by quantifying the production of major crops between 2011 and 2020. We then identified and analyzed the current problems in terms of policy support and other aspects. We used an integrated AHP-fuzzy evaluation method to evaluate the comprehensive benefits of different straw return amounts, and we determined the amount of straw that should be returned to the soil to produce the best comprehensive benefits. We provide suggestions for the current main problems regarding the amount of crop straw to return to the soil in Heilongjiang Province, arguing that choosing a reasonable straw return amount will help farmers increase profit, reduce environmental pollution, and contribute to the sustainable development of the environment.

**Keywords:** sustainable development of the environment; crop straw return volume; resource reuse; analytic hierarchy process; fuzzy comprehensive evaluation; performance evaluation



## 1. Introduction

Today, interest in the development of global renewable energy and clean energy uses is strong, and countries are increasingly concerned about environmental protection [1–3]. With advances in crop production technology and agricultural machinery, crop production has been increasing. However, as food production has increased, the excessive amount of straw resources has become an increasingly prominent problem [4]. A sustainable agricultural system is a method that is widely practiced by rural farming communities, where farmers grow various types of crops and raise livestock in the same area with the aim of being able to use land optimally, but mostly carried out conventionally [5,6]. The agricultural intensification system has not been implemented properly, for example, the use of fertilizers to increase crop yields, which is due to limited funds [7,8]. The use of local resources is one alternative to meet these limitations, namely by utilizing agricultural residue as a source of energy in the form of organic fertilizer/animal feed/animal bedding [9]. Many scholars have observed that straw return is beneficial to protecting the ecological environment of farmland, improving the physical and chemical properties of soil, and increasing the yield of crops [10]. To effectively and quickly prevent straw

burning and the air pollution it causes, straw return is gaining popularity in China because of its environmental friendliness and ease of implementation. Straw return can increase the organic carbon and nitrogen storage in the soil, thus increasing crop yields [11].

In this study, we evaluated the benefits of different straw return amounts and find the straw return amount with the highest combined benefits. We tentatively believe that the full amount of straw returned to the field may have the best benefit. As an agricultural by-product, straw has limited value on its own. When choosing the amount of straw to be returned to the field, choosing a suitable amount is difficult if only a single benefit is considered, which is not in line with the goal of sustainable development and is not conducive to the development of the practice of straw return. The comprehensive benefits of returning straw to the field need to be considered. We focused on the comprehensive evaluation of the economic, ecological, and social benefits of returning straw to the field. Future comparative studies can also be conducted for different straw return patterns.

Based on the characteristics of the agricultural structure in different areas in Heilongjiang Province and the need to promote sustainable agricultural development and improve the agricultural ecological environment, explorations of the comprehensive use technology of returning straw to the field are urgently needed so a suitable promotion system can be built. Research on the selection of different straw return quantities is also needed in Heilongjiang, and analyzing the quality of straw that is suitable for return in Heilongjiang Province is especially important to enable development suggestions to be developed and relevant choices to be made.

The innovations of this study can be summarized in the following two aspects.

(1) We determined the yields of maize, rice, and soybeans, and we estimated the amount of straw resources of these crops in Heilongjiang between 2011 and 2020. We also estimated the amount of straw resources of major crops in each region in Heilongjiang in 2020, thus providing a clearer understanding of the current situation and problems facing straw return in Heilongjiang Province.

(2) At present, most of the studies on returning straw to the field have focused on the effects of straw return on the physicochemical properties of the soil. Some scholars have studied the behavioral decisions of growers in straw return, but fewer studies have been conducted from the perspective of the comprehensive benefits of different amounts of straw returned to the field. In this study, we considered the ecological benefits of returning straw to the field, as well as its economic and social benefits. We also explored which amount of straw return produces the most comprehensive benefits by constructing a comprehensive benefit evaluation system. We also provide suggestions for promoting straw return, selecting the amount of rice straw to return to the field, as well as comprehensively enhancing the benefits of this practice.

## 2. Literature Review

### 2.1. Status of Research on Straw Return

Minomo et al. [12] found that straw burning produces large amounts of chlorides and releases them into the air. Chakraborty et al. [13] concluded that straw burning causes serious air pollution and affects human health, and that the particulate matter produced by burning irritates the eyes and causes them to water, irritates the nose and leads to poor breathing, and irritates the throat and leads to coughing and other diseases. Straw return was also found to have a strong effect on methane ($CH_4$) emissions from rice fields [14]. Long-term trials on wheat and corn straw in northern China proved that straw return plus nitrogen fertilizer can improve soil quality and increase yields [15]. The straw retained in agricultural fields can increase organic carbon and nitrogen stocks, thus potentially increasing crop yields [16]. Kaur et al. [17] found through their study that straw burning produces large amounts of chlorides and releases them into the air. Glab and Kulig [18] found that straw return improves soil aggregates. Straw return can enhance soil enzyme activity and increase soil fertility, improve crop yield, and thus increase farmers' income [19]. The straw return practice is also flourishing in developed areas such as Europe and the

United States. Humberto et al. [20] stated that not implementing a straw return policy would reduce soil fertility, so they proposed directly returning waste straw into the soil. Silalertruksa and Gheewala [21] explored straw resources and found that the use of straw as a fertilizer for soil fertilization by returning it to the field had the highest nutrient efficiency compared with its direct use as animal feed, energy, or industrial raw material. Ndzelu et al. [22] studied the effect of different corn stover return methods on the humus structure of the soil, concluding that corn stover return is a force driving the stabilization of carbon in the soil and is important for the development of sustainable agriculture. Straw contains large amounts of organic acids, which can be reused by decomposition in the return process to increase the nutrient reserves in the soil. The use of straw can also reduce the amount of fertilizer used in arable land, thus mitigating the effect of various environmental pollution problems caused by the direct burning of straw and the destruction of arable land [23]. Ren et al. [24] conducted a three-year straw return experiment to confirm the effect of different straw return practices on nitrogen transformation in fertilized soils. The results showed that continuous straw return is required to maintain the nitrogen retention function of the soil. Many efforts have been devoted worldwide to returning straw to the field.

### 2.2. Status of Research on Comprehensive Agricultural Benefit Evaluation

Many researchers have evaluated the comprehensive economic benefits of returning crop straw to the field. Conway [25] reported that a comprehensive economic efficiency evaluation index system should include productivity, sustainability, etc. Rasul and Thapa [26] reported that the current researchers have generally used the goal–guide–concept and pressure–state–response systems to construct systems for evaluating ecological agriculture. Van Cauwenbergh et al. [27] provided comprehensive assessment framework criteria and indicators for evaluating the sustainability of the U.S. agricultural system, commonly referred to as the SAFE framework. Wei et al. [28] constructed a comprehensive assessment model to improve the crop ecosystem and sustainable development in China and applied it to the wheat–maize cultivation system on the North China Plain. Feng et al. [29] constructed a comprehensive performance evaluation model of regional water resources in China based on the fuzzy optimization principle.

### 2.3. Status of Research on AHP

The analytic hierarchy process (AHP) was proposed by Saaty [30] as an effective method to deal with unquantifiable parameters and is used to quantify influencing factors. Mikhailov and Tsvetinov [31] used the fuzzy analytic hierarchy process to deal with uncertainty and imprecision in the service evaluation process. Gungor et al. [32] proposed a personnel selection system based on the fuzzy analytic hierarchy process to evaluate the best and most appropriate personnel for handling qualitative and quantitative criteria ratings. Chou et al. [33] used the fuzzy analytic hierarchy process to assess the weight of each criterion in science and technology human resources. The AHP is suitable for choices related to complex systems and for making choices among several alternatives [34–37]. The analytic hierarchy process can be used to efficiently analyze complex problems that cannot be precisely quantified. Its arithmetic process is simple, and the results are reliable, so the analytic hierarchy process is chosen as a weighting method.

### 2.4. Status of Research on Fuzzy Comprehensive Evaluation

Some environmental researchers developed an advanced assessment method based on fuzzy logic: the fuzzy comprehensive evaluation method [38,39]. Zadeh [40] proposed fuzzy set theory. However, the fuzzy comprehensive evaluation method became popular after the 1980s and is mainly used by researchers in electronics and electrical and computer engineering. Now, the method is being used in increasing numbers of studies. Fuzzy theory was designed to explain the uncertainty of the real situation. The fuzzy comprehensive evaluation method has also been used in the environmental field [41–43]. The fuzzy comprehensive evaluation method can effectively deal with problems concerning the

information of a fuzzy comprehensive evaluation object, has strong systematicity, can solve the problem caused by some of the fuzzy influencing factors being difficult to quantify, and can produce clearer results. It can fully use the information contained in the original index data to reflect the differences between programs, thus strengthening the scientific basis of the evaluation conclusion. Therefore, the fuzzy comprehensive evaluation method could be applied to evaluate the comprehensive benefits of different straw return patterns [44–47]. The fuzzy comprehensive evaluation method, as an effective method most commonly used in fuzzy decision making, exactly fits our needs to evaluate the comprehensive benefits of returning crop straw to the field.

## 3. Analysis of the Current Situation and Problems of Returning Straw to the Field in Heilongjiang Province

### 3.1. Analysis of the Yield of Major Crops in Heilongjiang Province

Northeast black soil is one of the three major black soils in the world. Heilongjiang is located in the core black soil area in Northeast China. The land contains sufficient organic matter. Heilongjiang also has sufficient natural resources and a healthy natural environment. Recently, the grain yield in Heilongjiang has been increasing each year. The province is not only self-sufficient, but it can also export grain to other provinces, helping to guarantee food security in China. Corn, rice, and soybeans are the main cash crops in Heilongjiang, and the sown area and yield are both gradually rising. However, the corresponding amounts of by-products, such as straw and miscellaneous materials, are also gradually rising. In 2020, Heilongjiang produced 75.41 million tons of grain, including 36.47 million tons of corn, 28.96 million tons of rice, and 9.2 million tons of soybeans, for a total output of 74.63 million tons of the three major crops, accounting for 98.96% of the total grain yield, as shown in Figure 1.

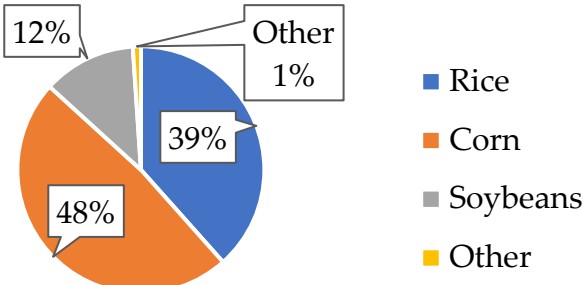

**Figure 1.** Major crops in Heilongjiang in 2020 as a percentage of yield.

The yields of major crops and total grain production in Heilongjiang from 2011 to 2020 are shown in Table 1. Overall, the total grain yield in Heilongjiang had trended upward from 2011 to 2020. Among the major crops, corn yield rose and then slightly fell; rice yield rose, then fell, and then rose again; and soybean yield showed a trend of falling and then rising.

**Table 1.** The main crop yield and total grain yield (million tons) between 2011 and 2020 in Heilongjiang Province.

| Year | Corn | Rice | Soybeans | Total Grain |
|------|------|------|----------|-------------|
| 2011 | 26.76 | 20.62 | 5.41 | 55.71 |
| 2012 | 28.88 | 21.71 | 4.63 | 57.61 |
| 2013 | 32.16 | 22.21 | 3.86 | 60.04 |
| 2014 | 33.43 | 22.51 | 4.60 | 62.42 |
| 2015 | 35.44 | 22.00 | 4.28 | 63.24 |
| 2016 | 31.27 | 22.55 | 5.62 | 60.59 |
| 2017 | 37.03 | 28.19 | 6.89 | 74.10 |
| 2018 | 39.82 | 26.86 | 6.57 | 75.07 |
| 2019 | 39.40 | 26.63 | 7.80 | 75.03 |
| 2020 | 36.47 | 28.96 | 9.20 | 75.41 |

According to Table 1, the change in the yield of major crops in Heilongjiang Province from 2011 to 2020 is plotted in Figure 2.

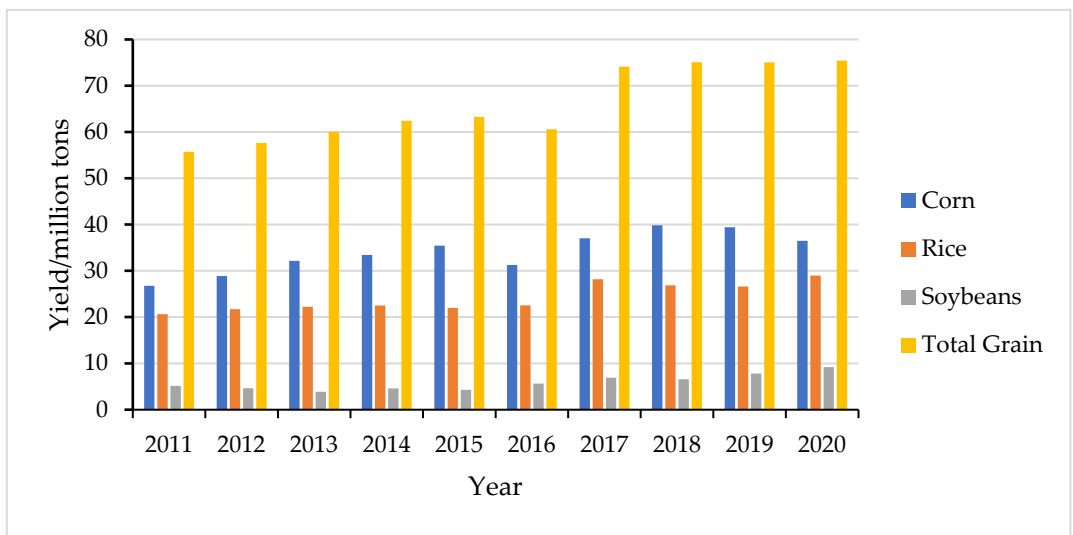

**Figure 2.** Changes in yield of major crops in Heilongjiang Province, 2011–2020.

Combining Table 1 and Figure 2, we found that between 2011 and 2020, the maize yield in Heilongjiang, except for a significant decrease in 2016, showed an overall upward trend. In 2016, the area of maize sowing was reduced by approximately 20 million mu in the province to implement pilot work (crop rotation with fallow) of the Ministry of Agriculture and the provincial government. In the ten-year period, the maximum corn yield in Heilongjiang appeared in 2018, at 39.82 million tons, an increase of 2.79 million tons over 2017. From 2011 to 2015, corn yield grew, and the yield increase was large, with a decline in yield in 2016. Corn yield then leveled off after a substantial increase in 2017, with a slight decline in 2020. Total rice yield has been steadily increasing from 2011 to 2017, with rice yield increasing by 7.57 million tons in 2017 compared with 2011. Rice yield declined from 2017 to 2019, with a decline of 1.56 million tons in 2019 compared with 2017, but with a rebound in 2020. Soybean yield showed an overall trend of declining and then rising; from 2010 to 2013, soybean yield declined each year, with the lowest yield occurring in 2013 at 3.86 million tons. From 2013 to 2020, soybean yield substantially increased, with the yield reaching 7.8 million tons in 2019, doubling that of 2013. The main reason is that in recent years, the state and Heilongjiang Province have subsidized soybean cultivation, which has motivated farmers to plant soybeans. As such, the area under soybean cultivation has continued to increase and the yield has risen. In the 10-year period, the total grain yield in Heilongjiang Province continually increased, except for 2016, with 2020 being the maximum total grain yield in Heilongjiang, reaching 75.41 million tons.

### 3.2. Estimation of Main Crop Straw Resources in Heilongjiang Province

3.2.1. Method of Measuring Amount of Straw Resources

The ratio of crop straw resources to crop yield is called the grass-to-grain ratio, and the method of calculating crop straw resources through the grass-to-grain ratio is called the grass-to-grain ratio method, which we used in this study to measure crop straw resources. The calculation formula is as follows:

$$R = M/M_p \tag{1}$$

where $R$ is the grass-to-grain ratio, $M$ is the amount of crop straw resources, and $M_p$ is the crop yield.

Crop straw resources include theoretical and collectible resources.

1. Theoretical resources

The theoretical resource amount of crop straw is calculated as follows:

$$P_y = F \times R \tag{2}$$

where $P_y$ is the theoretical resource of crop straw and $F$ is the crop yield.

2. Amount of collectible resources

The amount of collectable crop straw resources is calculated as:

$$P_j = P_y \times C \tag{3}$$

where $P_j$ is the amount of collectable crop straw resources and $C$ is the crop straw collectable coefficient.

3.2.2. Analysis of Theoretical Resource Measurement of Crop Straw in Heilongjiang

We calculated the theoretical resources of major crop straws in Heilongjiang by crop yield and crop grass-to-grain ratio. Crop yields were queried from the Heilongjiang Provincial Statistical Yearbook and the official website of the Ministry of Agriculture and Rural Affairs. We calculated the grass-to-grain ratio using the ratio recommended in the Notice of the General Office of the Ministry of Agriculture and Rural Affairs on the Construction of Crop Straw Resources Ledger, as shown in Table 2.

**Table 2.** Reference data of grass-to-grain ratio in different agricultural areas.

| Major Agricultural Areas | Corn | Rice | Soybeans |
|---|---|---|---|
| Northeast Agricultural Area | 1.86 | 0.97 | 1.70 |
| North China Agricultural Area | 1.73 | 0.93 | 1.57 |
| Middle and Lower Yangtze River Agricultural Area | 2.05 | 1.28 | 1.68 |
| Northwest Agricultural Area | 1.52 | / | 1.07 |
| Southwest Agricultural Area | 1.29 | 1.00 | 1.05 |
| Southern Agricultural Area | 1.32 | 1.06 | 1.08 |

Based on the grass-to-grain ratio coefficients for corn, rice, and soybeans and the yields of corn, rice, and soybeans in Heilongjiang Province between 2011 and 2020, given in the table, we calculated the theoretical resources of straw for corn, rice, and soybeans in Heilongjiang Province between 2011 and 2020, as shown in Table 3.

**Table 3.** Theoretical resources of major crop straws (million tons) between 2011 and 2020 in Heilongjiang Province.

| Year | Amount of Corn Straw | Amount of Rice Straw | Amount of Soybean Straw | Total Amount of Straw |
|---|---|---|---|---|
| 2011 | 49.77 | 20.00 | 9.20 | 78.97 |
| 2012 | 53.72 | 21.06 | 7.88 | 82.65 |
| 2013 | 59.83 | 21.54 | 6.57 | 87.94 |
| 2014 | 62.19 | 21.84 | 7.83 | 91.85 |
| 2015 | 65.92 | 21.34 | 7.28 | 94.54 |
| 2016 | 58.17 | 21.88 | 9.57 | 89.61 |
| 2017 | 68.88 | 27.35 | 11.72 | 107.95 |
| 2018 | 74.07 | 26.05 | 11.18 | 111.30 |
| 2019 | 73.28 | 25.84 | 13.27 | 112.39 |
| 2020 | 67.83 | 28.09 | 15.64 | 111.56 |

According to Table 3, the theoretical resource changes in the major crop straws in Heilongjiang Province from 2011 to 2020 were plotted (Figure 3).

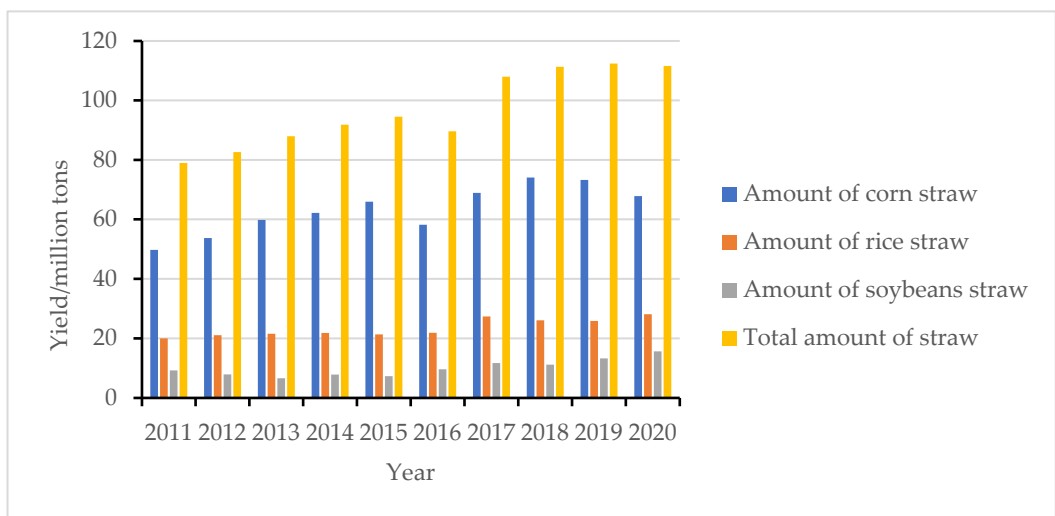

**Figure 3.** Changes in theoretical resources of major crop straws from 2011 to 2020 in Heilongjiang Province.

According to Table 3 and Figure 3, from an overall perspective, the theoretical straw resources of major crops in Heilongjiang were increasing, from 78.97 million tons in 2011 to 111.56 million tons in 2020, which is an increase of 41.27%. The theoretical straw resources steadily rose from 2011 to 2015, fell slightly in 2016, and continued to steadily rise from 2017 to 2019, with a slight fall in 2020.

Among the three main types of crop straw, the theoretical resources of corn straw experienced the largest growth, from 49.77 million tons in 2011 to 67.83 million tons in 2020, an increase of 18.06 million tons. Rice straw resources showed larger growth and growth rate, from 20 million tons in 2011 to 28.09 million tons in 2020, an increase of 40.45%. The increase in the theoretical resources of soybean straw has been the largest overall, with fluctuating trends from 2011 to 2018 and rapid growth in 2019 and 2020, from 9.2 million tons in 2010 to 15.64 million tons in 2020, an increase of 70%. In 2020, the theoretical resources of corn straw were the highest, accounting for 60.8% of the theoretical straw resources of the three major crops. Rice straw accounted for 25.18%, and soybean straw accounted for 14.02%.

3.2.3. Analysis of Collectable Crop Straw Resources in Heilongjiang Province

The theoretical collectable amount of crop straw is the actual collectable amount of straw, which we calculated by multiplying the theoretical resource amount of crop straw by the collection coefficient of crop straw. The collection coefficient of crop straw in this study, using the collection coefficient proposed in the document of the General Office of the Ministry of Agriculture and Rural Affairs about improving the construction of the crop straw resource ledger, is shown in Table 4. Because the agricultural mechanization rate in Heilongjiang Province is above 96%, the collection coefficients we selected in this study were 0.85 for corn straw, 0.74 for rice straw, and 0.56 for soybean straw.

**Table 4.** Straw collection coefficients.

| Straw Type | Leave Stubble Height (cm) | Collection Coefficient |
|---|---|---|
| Corn straw | Mechanical Harvest 15 | 0.85 |
| | Manual Harvest 7 | 0.90 |
| Rice straw | Mechanical Harvest 15 | 0.74 |
| | Manual Harvest 7 | 0.83 |
| Soybean straw | / | 0.56 |

Based on the theoretical resources of straw from the major crops in Heilongjiang Province derived from Table 3 and the straw collection coefficients of these major crops in the province, we calculated the collectable straw resources of the major crops in Heilongjiang Province from 2011 to 2020, as shown in Table 5.

**Table 5.** Collectable resources of major crop straws in Heilongjiang Province from 2011 to 2020 (million tons).

| Year | Collectable Resources of Corn Straw | Collectable Resources of Rice Straw | Collectable Resources of Soybean Straw | Total Resources |
|---|---|---|---|---|
| 2011 | 42.30 | 14.80 | 5.15 | 62.25 |
| 2012 | 45.66 | 15.58 | 4.41 | 65.65 |
| 2013 | 50.85 | 15.94 | 3.68 | 70.47 |
| 2014 | 52.85 | 16.16 | 4.38 | 73.39 |
| 2015 | 56.03 | 15.79 | 4.08 | 75.90 |
| 2016 | 49.44 | 16.19 | 5.36 | 70.99 |
| 2017 | 58.55 | 20.24 | 6.56 | 85.35 |
| 2018 | 62.96 | 19.28 | 6.26 | 88.50 |
| 2019 | 62.29 | 19.12 | 7.43 | 88.84 |
| 2020 | 57.66 | 20.79 | 8.66 | 87.11 |

Overall, the total amount of straw resources in Heilongjiang Province is huge. The collectable straw resources of the three major crops in Heilongjiang Province increased from 62.25 million tons in 2011 to 87.11 million tons in 2020, an increase of 1.4 times. During this period, the collectable corn straw resources increased from 42.3 million tons in 2011 to 57.66 million tons in 2020, an increase of 1.36 times; those of rice straw increased from 14.8 million tons in 2011 to 20.79 million tons in 2020, an increase of 1.41 times; and the collectable soybean straw resources increased from 5.15 million tons in 2011 to 8.66 million tons in 2020, an increase of 1.68 times.

### 3.3. Regional Distribution of Crop Straw Resources in Heilongjiang Province

To identify the spatial distribution of crop straw in Heilongjiang, we used the 2020 yield data of crops in thirteen major cities in Heilongjiang, as given in Table 6. The distribution of cities in Heilongjiang Province is shown in Figure 4. We measured the crop straw resources of each city in Heilongjiang Province in 2020 according to the calculation formulas of the grass-to-grain ratio and collectability coefficient mentioned, as shown in Table 7.

**Table 6.** Yield of major crops by region (million tons) of Heilongjiang Province in 2020.

| City | Corn Yield | Rice Yield | Soybean Yield | Total Yield |
|---|---|---|---|---|
| Harbin | 7.8418 | 3.8298 | 0.5051 | 12.2281 |
| Qiqihar | 7.0976 | 2.9608 | 1.5574 | 11.8204 |
| Jixi | 1.8427 | 3.6571 | 0.2874 | 5.7957 |
| Hegang | 0.7618 | 2.1731 | 0.2322 | 3.1724 |
| Shuangyashan | 1.9596 | 3.1101 | 0.5990 | 5.6751 |
| Daqing | 3.4722 | 0.8031 | 0.1609 | 4.5342 |
| Yichun | 0.2676 | 0.3468 | 0.2652 | 0.8857 |
| Jiamusi | 2.3893 | 7.7061 | 0.9361 | 11.0395 |
| Qitaihe | 0.7818 | 0.1510 | 0.0890 | 1.0233 |
| Mudanjiang | 2.1019 | 0.3246 | 0.4480 | 2.9684 |
| Heihe | 1.9579 | 0.1081 | 2.8395 | 5.1754 |
| Suihua | 7.4029 | 2.5588 | 0.9983 | 11.0814 |
| Daxinganling | 0.0091 | / | 0.2538 | 0.2751 |

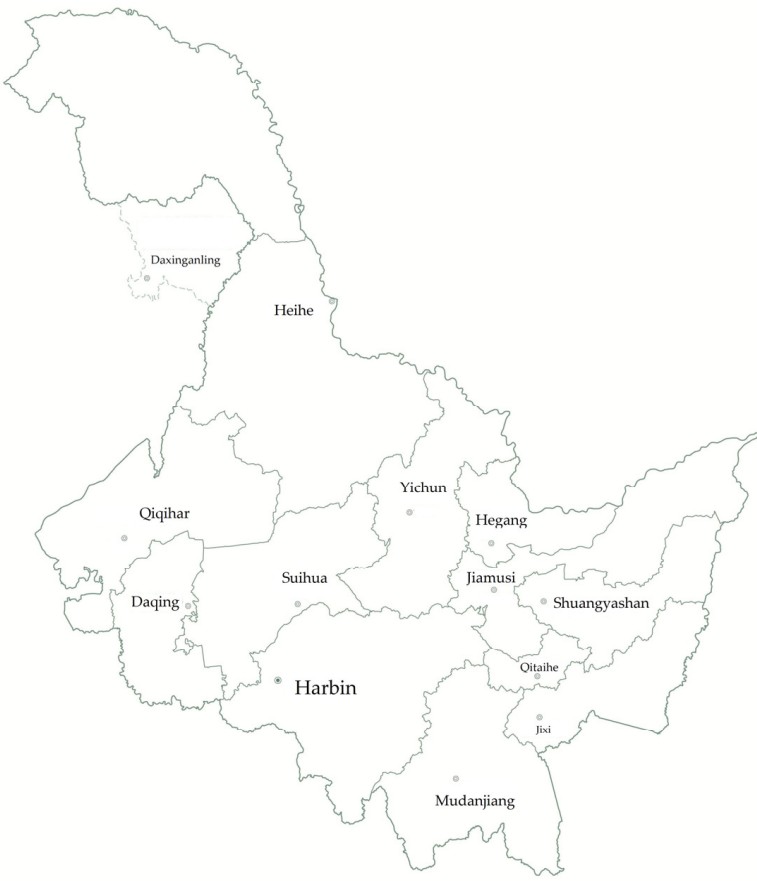

**Figure 4.** City map of Heilongjiang Province.

**Table 7.** The main crop straw resources by region (million tons) of Heilongjiang Province in 2020.

| City | Corn Straw | | Rice Straw | | Soybean Straw | | Total Amount of Straw | |
|---|---|---|---|---|---|---|---|---|
| | Theoretical Resources | Collectable Resources | Theoretical Resources | Collectable Resources | Theoretical Resources | Collectable Resources | Theoretical Resources | Collectable Resources |
| Harbin | 14.5857 | 12.3979 | 3.7149 | 2.7490 | 0.8587 | 0.4809 | 19.1593 | 15.6278 |
| Qiqihar | 13.2015 | 11.2213 | 2.8720 | 2.1253 | 2.6476 | 1.4826 | 18.7211 | 14.8292 |
| Jixi | 3.4274 | 2.9133 | 3.5474 | 2.6251 | 0.4886 | 0.2736 | 7.4634 | 5.8120 |
| Hegang | 1.4169 | 1.2044 | 2.1079 | 1.5599 | 0.3947 | 0.2211 | 3.9196 | 2.9853 |
| Shuangyashan | 3.6449 | 3.0981 | 3.0168 | 2.2324 | 1.0183 | 0.5702 | 7.6800 | 5.9008 |
| Daqing | 6.4583 | 5.4895 | 0.7790 | 0.5765 | 0.2735 | 0.1532 | 7.5108 | 6.2192 |
| Yichun | 0.4977 | 0.4231 | 0.3364 | 0.2489 | 0.4508 | 0.2525 | 1.2850 | 0.9245 |
| Jiamusi | 4.4441 | 3.7775 | 7.4749 | 5.5314 | 1.5914 | 0.8912 | 13.5104 | 10.2001 |
| Qitaihe | 1.4541 | 1.2360 | 0.1465 | 0.1084 | 0.1513 | 0.0847 | 1.7519 | 1.4291 |
| Mudanjiang | 3.9095 | 3.3231 | 0.3149 | 0.2330 | 0.7616 | 0.4265 | 4.9860 | 3.9826 |
| Heihe | 3.6417 | 3.0954 | 0.1049 | 0.0776 | 4.8272 | 2.7032 | 8.5737 | 5.8762 |
| Suihua | 13.7694 | 11.7040 | 2.4820 | 1.8367 | 1.6971 | 0.9504 | 17.9485 | 14.4911 |
| Daxinganling | 0.0169 | 0.0144 | / | / | 0.4315 | 0.2416 | 0.4484 | 0.2560 |

Table 7 shows that the distribution of straw resources in various cities in Hei-longjiang Province was uneven, with more crop straw resources located in the central and southern regions, which show higher potential for use. The straw resources in the four cities of Harbin, Qiqihar, Jiamusi, and Suihua were higher, having collectable straw resources of 15.6278, 14.8292, 10.201, and 14.4911 million tons, respectively. The four cities of Shuangyashan, Daqing, Jixi, and Heihe had average straw resources, with collectable straw resources of 5.908, 6.2192, 5.812, and 5.8762 million tons, respectively. Straw-resource-poor areas consisted of Mudanjiang, Hegang, Qitaihe, Yichun, and Daxinganling, having collectable straw resources of 3.982 million, 2.9853 million, 1.4291 million, 924,500, and 256,000 tons, respectively.

The distribution patterns of different crops differ throughout the province, mainly related to the climatic conditions, which affect where crops are grown throughout Heilongjiang Province. The corn stover resources are generally high in the south and low in the north, being relatively abundant in Harbin (12,397,900 tons), Qiqihar (11,221,300 tons), and Suihua (11,740,000 tons) in the south, for a total of 35,323,200 tons of corn stover resources in these three cities, accounting for more than half of the total corn stover resources in the province. In the northeast, Daxinganling (14,400 tons), Yichun (423,100 tons), Qitaihe (1,236,000 tons), Hegang (1,204,400 tons), and other areas have low amounts of collectable corn straw resources. These four areas have total corn straw collectable resources of 2,877,900 tons, accounting for only 5% of the province's corn straw resources. The overall amount of collectable rice straw resources showed a gradual decreasing trend from south to north: the northeastern city of Jiamusi (5,531,400 tons) had the largest resources; Harbin (2,740,900 tons), Jixi (2,625,100 tons), and Shuangyashan (2,223,240 tons) all had collectable rice straw resources of more than 2 million tons. These four cities accounted for more than half of the province's total. Due to the harsh climatic conditions in the northernmost part of the province, the Daxinganling area has a short frost-free period and insufficient accumulated temperature, which makes it unsuitable for growing rice. Therefore, the production of rice straw in the Daxinganling area was zero. The distribution of soybean straw resources was concentrated mainly in the city of Heihe, followed by the cities of Qiqihar, Suihua, and Jiamusi. In Heihe, Qiqihaer, Suihua, and Jiamusi, the soybean straw resources totaled 2.732 million, 1.4826 million, 950,400, and 891,200 tons, respectively. Heihe had the largest collectable soybean straw resources; Qiqihar, Suihua, and Jiamusi had medium resource levels. In other areas, the soybean straw resources were low, the least of which was 84,700 tons in Qitaihe, accounting for only 3.1% of that of Heihe.

*3.4. Status of Returning Straw to the Field in Heilongjiang Province*

Heilongjiang is a major grain-producing province, so straw production is also high in this area. With advances in science, straw resources have been researched extensively. Optimizing the use of straw resources can reduce air pollution, and straw return is an important step to revitalize the economic development of rural Heilongjiang and improve farmers' incomes and living standards. Farmers have different methods of using straw due to crop type, environmental climate, soil type, and other conditions.

In addition to returning crop straw to the fields, some straw is used as heating for farmers, some as feed for livestock, and some as chemical fuel or construction material. The use of straw widely varies in the different regions of Heilongjiang due to the different methods and degrees of economic development. According to the collected data, in 2015, about half of the straw resources in Heilongjiang were used for home heating, and less than 5% of the straw was returned to farmland, most of which is still being burned. Therefore, the use of straw resources in Heilongjiang is unreasonable. Before the ban on straw burning, most farmers still disposed of straw in the traditional way: most of the straw was burned directly in the fields or used as cooking fuel, and only a small portion was returned to the fields, causing serious resource waste and environmental problems.

The three methods of returning rice straw to the field in Heilongjiang are as follows. First, straw is directly crushed after harvesting, which is a method widely used in China, and the operation process is simple. The second is to return the straw fertilizer to the ground, as this involves the recycling of resources, thereby increasing soil fertility and producing a soil conservation effect. Third, the straw is stored in the ground. This method is also more convenient; the straw is fed to livestock and then returned to the ground, thus reducing the cost of animal feed. Most areas in Heilongjiang use direct return of straw to the field, where the straw is buried in the soil.

*3.5. Analysis of Straw Return Problems in Heilongjiang Province*

1.  Insufficient straw recycling publicity and low farmer recognition. In addition, the education level of farmers limits straw recycling. On the one hand, most farmers

in Heilongjiang province have no knowledge of returning straw to the field, do not know that there are many uses for straw besides burning, and still think that crop straw is a waste. On the other hand, most of the people who stay in rural farming in Heilongjiang Province are elderly, most of whom are not highly educated and do not know enough about the long-term benefits of returning straw to the fields, social benefits and environmental protection, and do not know enough about the hazards of open burning of straw.

2. The level of straw utilization technology is low, and it is difficult to promote the implementation of existing technology. Local governments spend a lot of manpower and resources on banning straw burning, but the publicity of straw utilization laws and regulations and straw utilization technology is not in place. At the same time, the rural people left behind have a low level of education and insufficient knowledge of straw comprehensive utilization, so it is very difficult to learn and implement the relevant straw utilization technology.

3. Improper use of machines and tools. Inefficient machine operation and insufficient cooperation between machines lead to straw waste.

4. Insufficient decomposition of straw due to temperature limitations. Heilongjiang Province is located in a cold region with little rainfall. Summer precipitation is mainly concentrated in June to August. Winter temperatures are extremely low, and straw decomposition substantially decreases from August to October, inhibiting the effect of returning straw to the field.

5. Rural labor shortage exacerbates straw burning. As returning straw to the field requires labor, the severe shortage of rural labor, as well as giving farmland to under-populated families, leads to nonideal straw return.

6. Auxiliary fertilizer use methods and techniques need to be improved. In the process of returning straw to the field, the use of fertilizer should be based on the actual situation; the improper amount of fertilizer can cause crop nutrient overload and other situations, which can waste money and cause losses.

7. Insufficient government support. Heilongjiang Province introduced a policy to ban straw burning in 2014. However, in the implementation process, a perfect supervision system and detailed financial assistance plans are lacking, and the effect the ban has been far from expectations.

8. Farmers cannot reasonably determine the amount of straw that should be returned to the field. Incorporating the appropriate amount of straw in the field can provide comprehensive benefits, producing a multiplier effect. If the farmers cannot reasonably choose the amount of straw to be returned to the field, straw resources may be wasted.

## 4. Evaluation Index System Construction and Evaluation Methods

### 4.1. Evaluation Index System Construction

Based on existing study results, and according to the principles of selecting a comprehensive benefit evaluation index of straw return volume, we selected 3 primary evaluation indices of economic, ecological, and social benefits, and 10 secondary evaluation indices, including fertilizer application reduction rate per mu and soil organic matter increase, according to the attributes of each index.

#### 4.1.1. Economic Efficiency Index $B_1$

Economic efficiency reflects the relationship between the results of straw return and the cost, which requires a certain amount of investment to obtain results. The better the result of straw return, the greater the economic benefit. Therefore, we selected four indicators (total output value per mu, net profit per mu, capital production and investment ratio, and total cost per mu of straw returned to the field) as the economic benefit evaluation indicators.

1. Total output value per mu $C_1$

The total output value per mu is the total value of crops produced from farmland per unit area.

2. Net profit per mu $C_2$

Net profit per mu is the balance of the total crop production value minus all costs invested in the planting and straw return process, reflecting the net crop yield.

3. Capital production and investment ratio $C_3$

The capital production and investment ratio is the ratio of the sum of all capital inputs and outputs of crop cultivation and straw return, which measures the level of return of different amounts of straw to the field.

4. Total cost per mu of straw returned to the field $C_4$

The total cost per mu of straw returned to the field is the sum of all costs invested in the process of returning straw to the field, reflecting all resources consumed in the process.

### 4.1.2. Ecological Efficiency Index $B_2$

Ecological efficiency reflects the impact on the ecological environment, where the higher the eco-efficiency, the more rational the use of natural resources and the stronger the protection of the environment, thus promoting the sustainable development of straw return and driving the development of the agricultural economy. Therefore, we choose four ecological benefit indicators: amount of straw returned to the field, straw return rate, chemical fertilizer reduction per mu, and soil organic matter increase.

1. Amount of straw returned to the field $C_5$

Rice straw can be returned to the field to ensure food security, and a suitable amount of straw can be returned to the field with half the effort. The amount of straw returned to the field has a significant impact on ecological benefits.

2. Straw return rate $C_6$

The reasonable use of straw involves the full use of space and resources, which leads to higher ecological benefits. The straw return rate reflects the degree to which straw is returned to the field.

3. Chemical fertilizer reduction per mu $C_7$

The excessive use of chemical fertilizers can damage the ecological environment, and returning straw to the soil can theoretically reduce the applied amount of chemical fertilizers. This indicator is the reduction in fertilizer use in the process of crop cultivation under different straw return modes.

4. Soil organic matter increase $C_8$

Soil organic matter refers to the various organic substances contained in the soil, which has a catalytic effect on crop growth, and straw returned to the field can theoretically increase the soil organic matter content. This index is the rate of increase in soil organic matter under different straw return amounts.

### 4.1.3. Social Efficiency Index $B_3$

In addition to providing an adequate food supply, crop cultivation plays an active role in other activities in the national economy. As an important component of agricultural production, crop farming can improve social benefits. Providing crop products to ensure food security and driving rural labor to ensure social stability are both social benefits of crop farming. We selected the labor force role and technology satisfaction as social benefit evaluation indicators.

1. Labor-force-driven role $C_9$

This indicator refers to the amount of labor input throughout the whole planting process. The current rural labor exodus problem is serious, and labor-led crop planting can effectively promote local economic and social development, which play an important role in improving social benefits.

2. Technology satisfaction $C_{10}$

Growers' satisfaction with straw return is an important social benefit indicator, which reflects the reliability and ease of promotion of the practice of straw return.

Combined with the hierarchical analysis method, we developed a three-level index system to evaluate the comprehensive benefits of returning straw to the field. The first level is the target level, that is, the comprehensive benefit of straw return volume; the second level is the guideline level, including three indicators of economic, ecological, and social benefit; the third level is the indicator level, including 10 evaluation indicators, such as fertilizer application reduction rate per mu, soil organic matter increase, etc. The specific indicator system is shown in Table 8.

**Table 8.** Comprehensive benefit evaluation index system of straw return volume.

| Target Level | Guideline Level | Bottom Level | Action Direction |
|---|---|---|---|
| Comprehensive benefits A | Economic efficiency B1 | Total output value per mu C1 | + |
| | | Net profit per mu C2 | + |
| | | Capital production and investment ratio C3 | + |
| | | Total cost per mu of straw returned to the field C4 | - |
| | Ecological efficiency B2 | Amount of straw returned to the field C5 | + |
| | | Straw return rate C6 | + |
| | | Chemical fertilizer reduction per mu $C_7$ | + |
| | | Soil organic matter increase $C_8$ | + |
| | Social efficiency $B_3$ | Labor-force-driven role C9 | + |
| | | Technology satisfaction C10 | + |

*4.2. Weight Calculation at Each Level Based on AHP*

The main steps of the AHP are described below [36,48–50].

4.2.1. Building the Hierarchy Model

Based on the comprehensive benefit evaluation index system of straw return volume that we constructed (Table 8), we divided the factors involved into three levels: target, guideline, and bottom levels. Finally, we identified 10 evaluation indicators.

4.2.2. Constructing the Comparison Matrix

By introducing the 1–9 scaling method described in Table 9, we constructed a comparison matrix by scoring the impact of experts on various evaluation indicators. Then, we estimated the specific weights of each level of evaluation indicators, integrated the calculations, and finally determined the specific weights of all evaluation indicators.

**Table 9.** Scaling method.

| Scale | Meaning |
|---|---|
| 1 | Equal scale between two decision elements |
| 3 | Moderate scale of one decision element compared to another decision element |
| 5 | Strong scale of one decision element compared to another decision element |
| 7 | Extreme scale of one decision element compared to another decision element |
| 9 | Absolute scale of one decision element compared to another decision element |
| 2, 4, 6, 8 | Intermediate levels of the above scale |

We constructed the comparison matrix as shown in Table 10.

**Table 10.** Comparison matrix.

| Scale | $X_1$ | $X_2$ | $X_3$ | ... | $X_j$ |
|---|---|---|---|---|---|
| $X_1$ | $X_{11}$ | $X_{12}$ | $X_{13}$ | ... | $X_{1j}$ |
| $X_2$ | $X_{21}$ | $X_{22}$ | $X_{23}$ | ... | $X_{2j}$ |
| $X_3$ | $X_{31}$ | $X_{32}$ | $X_{33}$ | ... | $X_{3j}$ |
| ... | ... | ... | ... | ... | ... |
| $X_i$ | $X_{i1}$ | $X_{i2}$ | $X_{i3}$ | ... | $X_{ij}$ |

where $X_{ij} > 0$, $X_{ij} = 1/X_{ji}$, and $X_{ii} = 1$.

4.2.3. Assigning Weights to Indicators and Consistency Test

We calculated the indicator weights Wi with the square root method; the calculation formula is as follows:

$$\overline{W_i} = \sqrt[n]{\prod_{j=1}^{n} X_{ij}} \tag{4}$$

$$W_i = \frac{\overline{W_i}}{\left(\sum_{i=1}^{n} \overline{W_i}\right)} \tag{5}$$

We calculated the largest eigenvalue of the comparison matrix $\lambda_{max}$. The calculation formula is as follows:

$$\lambda_{max} = \sum_{i=1}^{n} \frac{(AW)_i}{nW_i} \tag{6}$$

We performed a consistency test on the judgment matrix, where *CR* is the test coefficient. The calculation formula is as follows:

$$CR = \frac{CI}{RI} \tag{7}$$

*CI* is a consistency indicator; *n* is the number of judgment matrix order. The calculation formula is as follows:

$$CI = \frac{(\lambda_{max} - n)}{(n-1)} \tag{8}$$

We obtained the *RI* by taking the arithmetic mean after repeating the calculation of the eigenvalues of the random judgment matrix several times. The *RI* values are detailed in Table 11 for the judgment matrix of order 1–9.

**Table 11.** RI values for matrices of order 1–9.

| n | 1 | 2 | 3 | 4 | 5 | 6 | 7 | 8 | 9 |
|---|---|---|---|---|---|---|---|---|---|
| RI | 0.00 | 0.00 | 0.58 | 0.9 | 1.12 | 1.24 | 1.32 | 1.41 | 1.45 |

To check the subjective bias of a judgment matrix, Saaty proposed the consistency indicator (CI). CI = 0 indicates complete consistency; when CI is close to 0, consistency is satisfactory. The larger the CI, the more serious the inconsistency. As deviations in consistency may be caused by random reasons, when testing whether the consistency of a judgment matrix is satisfactory, the CI must also be compared with the random coherence index (RI) to obtain the test coefficient CR. Using the mean RI, if CR = CI/RI < 0.10 (that is, when CR is less than 0.10), the judgment matrix is acceptable. Otherwise, the initially established judgment matrix is unsatisfactory and needs to be recalculated [37].

*4.3. Fuzzy Comprehensive Evaluation of Benefits*

Various methods can be used to return crop straw to the field, and different crops have different characteristics, so the benefits produced by them are uncertain. Additionally, other factors may affect the benefit evaluation process, which highlights that the comprehensive

evaluation of the benefit of returning crop straw to the field is a complex process, involving variables that are uncertain [38]. The main steps of the method are described next [44–47].

### 4.3.1. Establishment of Straw Return Evaluation Factor Set

The evaluation factor set U is a collection of evaluation factors. In the straw return evaluation system, the evaluation factor set U contains three evaluation factors in the first-level index, U = {$U_1$, $U_2$, $U_3$}, where $U_i$(i = 1, 2, 3) denotes the economic, ecological, and social benefit evaluation factors, respectively. After establishing the evaluation factor sets of the primary indicators, we establish the evaluation factor sets of the secondary indicators corresponding to each primary indicator.

1.  Economic benefits $U_1$ = {$U_{11}$, $U_{12}$, $U_{13}$, $U_{14}$}, which represent total output value per mu, net profit per mu, capital production and investment ratio, and total cost per mu of straw returned to the field, respectively.
2.  Ecological benefits $U_2$ = {$U_{21}$, $U_{22}$, $U_{23}$, $U_{24}$}, which represent the amount of straw returned to the field, the rate of straw returned to the field, the rate of chemical fertilizer reduction per mu, and the rate of increase of soil organic matter, respectively.
3.  Social benefits $U_3$ = {$U_{31}$, $U_{32}$}, indicating labor force driving effect and technology satisfaction, respectively.

### 4.3.2. Establishing Evaluation Scale Sets

The set of evaluation scales forms the rating scale used for scoring. We establish V = {$v_1$, $v_2$, $v_3$, $v_4$} as the set of evaluation scales by selecting the amount of straw returned to the field. The set is of the form {Excellent, Good, Medium, Poor}.

### 4.3.3. Determining Weights of Evaluation Indicators

In an evaluation system, different factors have different influences on the evaluation object. Therefore, in an actual evaluation process, the importance of each factor to the evaluation object is expressed by assigning a certain weight to each factor. We determined the weight of each factor in this study by the AHP method. The detailed process is shown in Section 5, and the results are shown in Table 16.

### 4.3.4. Establishing Single-Factor Fuzzy Evaluation

To evaluate a single indicator, we determine the affiliation rate of that evaluation indicator for the set of evaluation factors U and establish a fuzzy matrix vector. For example, 10 experts form a team to evaluate an indicator of the amount of straw returned to the field, and 5, 3, 2, and 0 of the experts evaluate it as excellent, good, average, and poor, respectively. Then, the fuzzy evaluation vector of the indicator of the amount of straw returned to the field is [0.5, 0.3, 0.2, 0] (the meaning of the vector is the percentage of votes of the experts who score the amount of straw returned to the field for each evaluation level). The vector satisfies the normalization.

### 4.3.5. Building a Fuzzy Factor Matrix

After establishing the single-factor fuzzy set, each indicator is evaluated to determine the fuzzy matrix vector and establish the fuzzy relationship matrix:

$$\tilde{R} = \begin{bmatrix} r_{11} & r_{12} & \cdots & r_{1n} \\ r_{21} & r_{22} & \cdots & r_{2n} \\ \cdots & \cdots & \cdots & \cdots \\ r_{n1} & r_{n2} & \cdots & r_{nn} \end{bmatrix} \tag{9}$$

where $r_{ij}$ is the percentage of expert votes for the evaluation score grade of $v_i$ for the ith evaluation factor $U_i$.

#### 4.3.6. Level 1 Fuzzy Comprehensive Evaluation

Let the indicator weight of $U_i$ be $A_i = (a_{i1}, a_{i2}, \cdots a_{in})$. We obtain the first-level fuzzy comprehensive evaluation as:

$$A_i * R_i = B_i (i = 1, 2, \cdots m) \tag{10}$$

#### 4.3.7. Level 2 Fuzzy Comprehensive Evaluation

Treating each $U_i$ as an element and $B_i$ as its single-factor judgment, the total judgment matrix is obtained as follows.

$$R = \begin{bmatrix} B_1 \\ B_2 \\ \vdots \\ B_k \end{bmatrix} \tag{11}$$

Let the weight of $U = \{U_1, U_2, U_3\}$ be $A = (a_1, a_2, \cdots a_k)$. The second-level fuzzy comprehensive evaluation is obtained as:

$$X = A * R \tag{12}$$

#### 4.3.8. Analysis of Evaluation Results

According to the calculation, the result X of the fuzzy comprehensive evaluation of a straw return quantity is a fuzzy vector. Let the rank score $Y = [1, 2, 3, 4]^T$, and we obtain the evaluation score $Z = X \times Y$. After calculating the fuzzy comprehensive evaluation results of all straw return quantities, we compare the scores, where the higher the score, the better the comprehensive benefit.

### 5. Example Analysis and Evaluation Results

#### 5.1. Weight Value Calculation and Consistency Test

The importance of each index was understood through expert scoring, and we constructed a two-by-two judgment matrix to calculate and determine the weight value of each index. The expert scoring in this study was achieved with questionnaires; we invited eight experts from various fields, such as university, enterprise, grassroots technical, and agricultural management fields, to complete these questionnaires. After the final confirmation of the experts, we constructed the judgment matrix according to the final scoring.

According to the index weight calculation method described in Section 4, we used SPSSPRO software to determine the first- and second-level index weights and to perform the consistency tests. The guideline-level A–Bi matrix and the results of the operation are shown in Table 12.

**Table 12.** Matrix and calculation results of weights of first-level indicators.

| A | $B_1$ | $B_2$ | $B_3$ | W | CR |
|---|---|---|---|---|---|
| $B_1$ | 1.0000 | 2.0000 | 2.0000 | 0.4934 | |
| $B_2$ | 0.5000 | 1.0000 | 0.5000 | 0.3108 | 0.0268 |
| $B_3$ | 0.5000 | 0.5000 | 1.0000 | 0.1958 | |

The bottom-level $B_1$–$C_i$ matrix and the results of the operations are shown in Table 13.

**Table 13.** Matrix and calculation results of economic efficiency index weights.

| $B_1$ | $C_1$ | $C_2$ | $C_3$ | $C_4$ | $W_1$ | CR |
|---|---|---|---|---|---|---|
| $C_1$ | 1.0000 | 0.2500 | 0.2500 | 1.0000 | 0.1030 | |
| $C_2$ | 4.0000 | 1.0000 | 0.5000 | 3.0000 | 0.3223 | |
| $C_3$ | 4.0000 | 2.0000 | 1.0000 | 3.0000 | 0.4558 | 0.0237 |
| $C_4$ | 1.0000 | 0.3333 | 0.3333 | 1.0000 | 0.1189 | |

The bottom-level $B_2$–$C_i$ matrix and the results of the operations are shown in Table 14.

**Table 14.** Matrix and calculation results of ecological efficiency index weights.

| $B_2$ | $C_5$ | $C_6$ | $C_7$ | $C_8$ | $W_2$ | CR |
|---|---|---|---|---|---|---|
| $C_5$ | 1.0000 | 3.0000 | 2.0000 | 0.3333 | 0.2616 | |
| $C_6$ | 0.3333 | 1.0000 | 2.0000 | 0.5000 | 0.1671 | 0.0869 |
| $C_7$ | 0.5000 | 0.5000 | 1.0000 | 0.3000 | 0.1182 | |
| $C_8$ | 3.0000 | 2.0000 | 3.0000 | 1.0000 | 0.4531 | |

The bottom-level $B_3$–$C_i$ matrix and the results of the operations are shown in Table 15.

**Table 15.** Matrix and calculation results of social efficiency index weights.

| $B_3$ | $C_9$ | $C_{10}$ | $W_3$ |
|---|---|---|---|
| $C_9$ | 1.0000 | 2.0000 | 0.6667 |
| $C_{10}$ | 0.5000 | 1.0000 | 0.3333 |

Through the calculation of the judgment matrix of the above primary and secondary index weights and the consistency test, we finally obtained the weight set W of the comprehensive benefit evaluation index, including $W_1$, $W_2$, and $W_3$ of the economic, ecological, and social benefit evaluation indices, respectively, for the rice straw return model.

W = [0.4934, 0.3108, 0.1958];
$W_1$ = [0.1030, 0.3223, 0.4558, 0.1189];
$W_2$ = [0.2616, 0.1671, 0.1182, 0.4531];
$W_3$ = [0.6667, 0.3333].

By calculation, we obtained the weight distribution of the combined indicators, as shown in Table 16.

**Table 16.** Distribution of index weights.

| Target Level | Weights | Guideline Level | Weights | Bottom Level | Weights |
|---|---|---|---|---|---|
| Comprehensive benefits A | 1 | Economic efficiency B1 | 0.4934 | Total output value per mu C1 | 0.0508 |
| | | | | Net profit per mu C2 | 0.1590 |
| | | | | Capital production and investment ratio C3 | 0.2249 |
| | | | | Total cost per mu of straw returned to the field C4 | 0.0587 |
| | | Ecological efficiency B2 | 0.3108 | Amount of straw returned to the field C5 | 0.0813 |
| | | | | Straw return rate C6 | 0.0519 |
| | | | | Chemical fertilizer reduction per mu C7 | 0.0367 |
| | | | | Soil organic matter increase $C_8$ | 0.1408 |
| | | Social efficiency B3 | 0.1958 | Labor-force-driven role C9 | 0.1305 |
| | | | | Technology satisfaction C10 | 0.0653 |

According to the weighting results, the weights of economic, ecological, and social efficiency in the primary index were 0.4934, 0.3108, and 0.1958, respectively. By comparing the weights of the three types of benefits, we concluded that economic efficiency most strongly influenced the comprehensive benefits, followed by ecological efficiency, and then social efficiency.

*5.2. Comprehensive Benefit Evaluation Using Fuzzy Comprehensive Evaluation Method*

According to the derived index weights, we invited 10 experts in the field of straw research from Jilin University and Northeast Agricultural University to score and calculate the benefits of straw return according to the fuzzy comprehensive integrated evaluation method, using YAANP software to obtain the comprehensive benefit scores and ranking of different straw return modes in Heilongjiang Province. In the specific analysis, three

aspects (economic, ecological, and social benefits) of different straw return modes were first evaluated. Then, the overall evaluations of the comprehensive benefits of different straw return modes were compared.

5.2.1. Comprehensive Benefit Evaluation of Returning the Full Amount of Straw to the Field

1.    Assign weights of indicators at each level

$W = \{0.4934, 0.3108, 0.1958\}$;
$W_{B1} = \{0.1030, 0.3223, 0.4558, 0.1190\}$;
$W_{B2} = \{0.2617, 0.1670, 0.1181, 0.4532\}$;
$W_{B3} = \{0.6665, 0.3335\}$.

2.    First-level fuzzy comprehensive evaluation

After processing the questionnaire data, the fuzzy matrix (affiliation matrix) corresponding to the economic benefits of returning the full amount of straw to the field, and the calculated results were obtained, as shown in Table 17.

**Table 17.** Fuzzy matrix of economic benefits of returning the full amount of straw to the field.

| $B_1$ | Poor | Medium | Good | Excellent |
|---|---|---|---|---|
| $C_1$ | 0.3000 | 0.1000 | 0.2000 | 0.4000 |
| $C_2$ | 0.0000 | 0.5000 | 0.5000 | 0.0000 |
| $C_3$ | 0.1000 | 0.2000 | 0.2000 | 0.5000 |
| $C_4$ | 0.1000 | 0.2000 | 0.4000 | 0.3000 |

The fuzzy matrix (affiliation matrix) corresponding to the ecological benefits of returning the full amount of straw to the field was obtained, and the calculated results are shown in Table 18.

**Table 18.** Fuzzy matrix of ecological benefits of returning the full amount of straw to the field.

| $B_2$ | Poor | Medium | Good | Excellent |
|---|---|---|---|---|
| $C_5$ | 0.3000 | 0.2000 | 0.2000 | 0.3000 |
| $C_6$ | 0.1000 | 0.1000 | 0.4000 | 0.4000 |
| $C_7$ | 0.1000 | 0.3000 | 0.2000 | 0.4000 |
| $C_8$ | 0.1000 | 0.2000 | 0.2000 | 0.5000 |

The fuzzy matrix (affiliation matrix) corresponding to the social benefits of returning the full amount of straw to the field was obtained, and the calculated results are shown in Table 19.

**Table 19.** Fuzzy matrix of social benefits of returning the full amount of straw to the field.

| $B_3$ | Poor | Medium | Good | Excellent |
|---|---|---|---|---|
| $C_9$ | 0.2000 | 0.3000 | 0.3000 | 0.2000 |
| $C_{10}$ | 0.2000 | 0.3000 | 0.1000 | 0.4000 |

Then, the first-level fuzzy comprehensive evaluation was carried out, so that the rank score $Y = [1, 2, 3, 4]^T$, $X_1 = W_{B1} \times R_{B1} = [0.0884, 0.2864, 0.3205, 0.3048]$, and the economic benefit evaluation score $Z_1 = X_1 \times Y = 2.8417$; $X_2 = W_{B2} \times R_{B2} = [0.1523, 0.1951, 0.2334, 0.4192]$, and we get the ecological benefit evaluation score $Z_2 = X_2 \times Y = 2.9194$; $X_3 = W_{B3} \times R_{B3} = [0.2000, 0.3000, 0.2333, 0.2667]$, and we get the social benefit evaluation score $Z_3 = X_3 \times Y = 2.5667$. From the evaluation score, we can see that the ecological benefit of returning the full amount of straw to the field is the best, the economic benefit is the second, and the social benefit is the lowest.

3.  Second-level fuzzy comprehensive evaluation

The fuzzy matrix (affiliation matrix) corresponding to the comprehensive benefits of returning full amount of straw to the field and the calculation results were obtained, as shown in Table 20.

**Table 20.** Fuzzy matrix of comprehensive benefits of returning full amount of straw to the field.

| A | Poor | Medium | Good | Excellent |
|:---:|:---:|:---:|:---:|:---:|
| $B_1$ | 0.0884 | 0.2864 | 0.3205 | 0.3048 |
| $B_2$ | 0.1523 | 0.1951 | 0.2334 | 0.4192 |
| $B_3$ | 0.2000 | 0.3000 | 0.2333 | 0.2667 |

Finally, the second-level fuzzy comprehensive evaluation was carried out, $X = W_A \times R_A$ = [0.1160, 0.2645, 0.2894, 0.3301], so that the rank score $Y = [1, 2, 3, 4]^T$, and the comprehensive benefit evaluation score $Z = X \times Y = 2.8336$ for returning the full amount of straw to the field was obtained.

5.2.2. Comprehensive Benefit Evaluation of Returning Half the Straw Volume to the Field

1.  Assign weights of the indicators at each level

$W = \{0.4934, 0.3108, 0.1958\}$;
$W_{B1} = \{0.1030, 0.3223, 0.4558, 0.1190\}$;
$W_{B2} = \{0.2617, 0.1670, 0.1181, 0.4532\}$;
$W_{B3} = \{0.6665, 0.3335\}$.

2.  First-level fuzzy comprehensive evaluation

By processing the questionnaire data, we obtained the fuzzy matrix (affiliation matrix) corresponding to the economic benefits of returning half the straw volume to the field. The calculated results are shown in Table 21.

**Table 21.** Fuzzy matrix of economic benefits of returning half the straw volume to the field.

| $B_1$ | Poor | Medium | Good | Excellent |
|:---:|:---:|:---:|:---:|:---:|
| $C_1$ | 0.3000 | 0.1000 | 0.2000 | 0.4000 |
| $C_2$ | 0.0000 | 0.3000 | 0.7000 | 0.0000 |
| $C_3$ | 0.3000 | 0.3000 | 0.2000 | 0.3000 |
| $C_4$ | 0.0000 | 0.4000 | 0.6000 | 0.0000 |

We obtained the fuzzy matrix (affiliation matrix) corresponding to the ecological benefits of returning half the straw volume to the field; the calculation results are shown in Table 22.

**Table 22.** Fuzzy matrix of ecological benefits of returning half the straw volume to the field.

| $B_2$ | Poor | Medium | Good | Excellent |
|:---:|:---:|:---:|:---:|:---:|
| $C_5$ | 0.2000 | 0.3000 | 0.2000 | 0.3000 |
| $C_6$ | 0.1000 | 0.2000 | 0.6000 | 0.1000 |
| $C_7$ | 0.2000 | 0.3000 | 0.1000 | 0.4000 |
| $C_8$ | 0.1000 | 0.2000 | 0.3000 | 0.4000 |

We obtained the fuzzy matrix (affiliation matrix) corresponding to the social benefits of returning half the straw volume to the field, with the calculated results shown in Table 23.

**Table 23.** Fuzzy matrix of social benefits of returning half the straw volume to the field.

| $B_3$ | Poor | Medium | Good | Excellent |
|---|---|---|---|---|
| $C_9$ | 0.1000 | 0.6000 | 0.3000 | 0.0000 |
| $C_{10}$ | 0.3000 | 0.4000 | 0.0000 | 0.3000 |

Then, we performed the first-level fuzzy comprehensive evaluation, $X_1 = W_{B1} \times R_{B1} =$ [0.1221, 0.2913, 0.4087, 0.1779], to obtain the economic benefit evaluation score $Z_1 = X_1 \times Y$ = 2.6425; $X_2 = W_{B2} \times R_{B2}$ = [0.1380, 0.2380, 0.3003, 0.3237] to obtain the ecological benefit evaluation score $Z_2 = X_2 \times Y$ = 2.8098; $X_3 = W_{B3} \times R_{B3}$ = [0.1667, 0.5333, 0.1999, 0.1001] to get the evaluation score of social benefit $Z_3 = X_3 \times Y$ = 2.2334. From the evaluation score, we found that the ecological benefit of returning half the straw volume to the field is the highest, followed by the economic benefit, and then the social benefit.

3. Second-level fuzzy comprehensive evaluation

We obtained the fuzzy matrix (affiliation matrix) corresponding to the comprehensive benefits of returning half the straw volume to the field and the calculation results, as shown in Table 24.

**Table 24.** Fuzzy matrix of integrated benefits of returning half the straw volume to the field.

| A | Poor | Medium | Good | Excellent |
|---|---|---|---|---|
| $B_1$ | 0.1221 | 0.2913 | 0.4087 | 0.1779 |
| $B_2$ | 0.1380 | 0.2380 | 0.3003 | 0.3237 |
| $B_3$ | 0.1667 | 0.5333 | 0.1999 | 0.1001 |

Finally, we conducted the second-level fuzzy comprehensive evaluation, $X = W_A \times R_A$ = [0.1306, 0.3022, 0.3599, 0.2072], so that the rank score $Y = [1, 2, 3, 4]^T$, and the comprehensive benefit evaluation score $Z = X \times Y$ = 2.6438 for returning half the straw volume to the field.

5.2.3. Comprehensive Benefit Evaluation of Not Returning Any Straw to the Field

1. Assign weights of the indicators at each level

$W$ = {0.4934, 0.3108, 0.1958};
$W_{B1}$ = {0.1030, 0.3223, 0.4558, 0.1190};
$W_{B2}$ = {0.2617, 0.1670, 0.1181, 0.4532};
$W_{B3}$ = {0.6665, 0.3335}.

2. First-level fuzzy comprehensive evaluation

By processing the questionnaire data, we obtained the fuzzy matrix (affiliation matrix) corresponding to the economic benefits of not returning straw to the field, and the calculated results are shown in Table 25.

**Table 25.** Fuzzy matrix of economic benefits of not returning straw to the field.

| $B_1$ | Poor | Medium | Good | Excellent |
|---|---|---|---|---|
| $C_1$ | 0.0000 | 0.3000 | 0.5000 | 0.2000 |
| $C_2$ | 0.0000 | 0.4000 | 0.5000 | 0.1000 |
| $C_3$ | 0.3000 | 0.2000 | 0.2000 | 0.3000 |
| $C_4$ | 0.1000 | 0.5000 | 0.4000 | 0.0000 |

We obtained the fuzzy matrix (affiliation matrix) corresponding to the ecological benefits of not returning straw to the field, and the calculation results are shown in Table 26.

**Table 26.** Fuzzy matrix of ecological benefits of not returning straw to the field.

| $B_2$ | Poor | Medium | Good | Excellent |
|---|---|---|---|---|
| $C_5$ | 0.4000 | 0.1000 | 0.5000 | 0.0000 |
| $C_6$ | 0.1000 | 0.2000 | 0.5000 | 0.2000 |
| $C_7$ | 0.3000 | 0.2000 | 0.1000 | 0.4000 |
| $C_8$ | 0.2000 | 0.3000 | 0.1000 | 0.4000 |

The fuzzy matrix (affiliation matrix) corresponding to the social benefits of not returning straw to the field was obtained, and the calculated results are shown in Table 27.

**Table 27.** Fuzzy matrix of social benefits of not returning straw to the field.

| $B_3$ | Poor | Medium | Good | Excellent |
|---|---|---|---|---|
| $C_9$ | 0.1000 | 0.6000 | 0.3000 | 0.0000 |
| $C_{10}$ | 0.4000 | 0.3000 | 0.0000 | 0.3000 |

Then, we performed the first-level fuzzy comprehensive evaluation, $X_1 = W_{B1} \times R_{B1}$ = [0.1486, 0.3104, 0.3514, 0.1896], and the economic benefit evaluation score $Z_1 = X_1 \times Y$ = 2.5818; $X_2 = W_{B2} \times R_{B2}$ = [0.2474, 0.2192, 0.2715, 0.2619], and the ecological benefit evaluation score was obtained $Z_2 = X_2 \times Y$ = 2.5479; $X_3 = W_{B3} \times R_{B3}$ = [0.2001, 0.4999, 0.1999, 0.1001] to get the evaluation score of social benefit $Z_3 = X_3 \times Y$ = 2.2000. From the evaluation score, we found that the economic benefit of not returning straw to the field was the highest, followed by the ecological, and then the social benefit.

3.  Second-level fuzzy comprehensive evaluation

We obtained the fuzzy matrix (affiliation matrix) corresponding to the comprehensive benefits of not returning straw to the field; the calculation results are shown in Table 28.

**Table 28.** Fuzzy matrix of integrated benefits of not returning straw to the field.

| A | Poor | Medium | Good | Excellent |
|---|---|---|---|---|
| $B_1$ | 0.1486 | 0.3104 | 0.3514 | 0.1896 |
| $B_2$ | 0.2474 | 0.2192 | 0.2715 | 0.2619 |
| $B_3$ | 0.2001 | 0.4999 | 0.1999 | 0.1001 |

Finally, we performed the second-level fuzzy comprehensive evaluation, $X = W_A \times R_A$ = [0.1791, 0.3063, 0.3156, 0.1990], so that the rank score $Y = [1, 2, 3, 4]^T$, and the comprehensive benefit evaluation score $Z = X \times Y$ = 2.5345 for not returning straw to the field.

5.2.4. Evaluation of Comprehensive Benefits of Different Straw Return Volumes

Based on the calculated benefit scores, we obtained the benefit evaluation results for different straw return volumes, as shown in Table 29.

**Table 29.** Benefit evaluation results of returning different straw volumes.

| Amount of Straw Return | Economic Benefit Score | Ecological Benefit Score | Social Benefit Score | Comprehensive Benefit Score |
|---|---|---|---|---|
| Full straw return | 2.8417 | 2.9194 | 2.5667 | 2.8336 |
| Half straw return | 2.6425 | 2.8098 | 2.2334 | 2.6438 |
| No straw return | 2.5818 | 2.5479 | 2.2000 | 2.5345 |

In terms of economic benefit results, the priority order of benefits for returning different straw amounts was full straw return > half straw return > no straw return. Returning straw to the field can indirectly increase the output value of crops, thus increasing the profit of

farmers, and returning either the full or half amount of the straw to the field can produce economic benefits.

In terms of ecological benefit results, the ranking of benefits for different straw return amounts is full straw return > half straw return > no straw return. Theoretically, returning straw to the field can reduce the input of chemical fertilizers, increase the soil organic matter content, and reduce the environmental pollution caused by burning straw. As such, returning the full and half amounts of straw to the field produces ecological benefits.

In terms of social benefit results, the ranking of the benefits of different straw return amounts is full straw return > half straw return > no straw return. Returning the full amount of straw to the field is labor-intensive, so can more effectively increase the income of many, creating a labor-driven effect as well as social satisfaction. Returning either the full or half amount of straw to the field produces social benefits.

According to the calculations to evaluate the comprehensive benefit of straw return, the ranking of the comprehensive benefits of returning different straw amounts to the field is full amount > half amount > no straw return. The overall benefit of returning the full amount of straw to the field is the highest, with an overall benefit evaluation score of 2.8336, followed by that of returning half the amount of straw, with an overall benefit evaluation score of 2.6438. The lowest overall benefit evaluation score of 2.5345 was obtained for returning no straw to the field. Although the benefit of returning half of the straw to the field is lower than that of returning the full amount of straw, this practice is less labor- and cost-intensive, so can be promoted according to the actual situation of growers on a trial basis. In terms of not returning straw to the field, although this practice saves the cost of returning straw to the field, the later processing of straw requires more capital and material inputs; otherwise, the straw will damage the ecological environment, and it does not produce comprehensive advantages over time. As such, the full or half amount of straw should be returned to the field as soon as possible.

*5.3. Suggestions for Response*

By analyzing the data on the amount of straw resources of the major crops in Heilongjiang Province in 2011–2020, including their distribution and use status, we found that, at present, although the popularity of the practice of returning straw to the field is high in Heilongjiang, the publicity on straw recycling is insufficient, the recognition by farmers of this practice is low, machines and tools are improperly used, straw decomposition is insufficient due to temperature restrictions, the shortage of rural labor is exacerbating straw burning, and government support of farmers is insufficient. Another problem is the inability to determine the appropriate amount of straw to be returned to the field. Given these problems and considering the results of the comprehensive benefit evaluation of returning different amounts of straw to the field, several aspects can be improved to further promote the development of straw return in Heilongjiang Province and ensure demands are met, the economy is promoted, straw is appropriately used, the environment is protected, and emissions are reduced.

(1)   Strengthen the publicity of returning straw to the field

Using a variety of current means of publicity, the practice of returning straw to the fields should be publicized to improve farmers' awareness of the method, correct the traditional thinking of farmers about straw burning, and establish a "green water and green mountains is the silver mountain" concept of environmental protection, so that farmers are deeply aware that straw is a resource and not waste, so as to improve the compliance of farmers with the straw-burning ban, thereby promoting the rational use of straw resources.

(2)   Strengthen the professional training of farmers

Through professional training, farmers should be provided with knowledge related to returning straw to the field, so farmers understand the hazards of open burning of straw, the ecological and economic benefits generated by returning straw to the field, and the

economic returns produced by this practice, so that farmers actively participate in returning straw to the field.

(3)  Pilot and demonstration work of returning straw to the field according to local conditions

Therefore, farmers more intuitively understand of the advantages of returning straw to the field, and province-wide pilot and demonstration projects of this practice should be set up according to local conditions, the different agricultural production characteristics in different areas, in line with the actual local test demonstration base. Through the pilot demonstration projects, farmers' ideologies can be changed to motivate them to return straw to the field.

(4)  Actively introduce advanced straw return technology at home and abroad

Combining the current situation of returning straw to the field in Heilongjiang and the objective demand of enterprises to return straw to the field, we can fully study and learn from the advanced experience and technology of advanced straw return at home and abroad and introduce these technologies to Heilongjiang. Considering the special geographical location of Heilongjiang Province, the alpine weather conditions, and the distribution of straw resources, we should perform a targeted transformation and strive to improve the rate of straw return in the province.

(5)  Encourage the research and development and promotion of straw return technology

Scientific research should be strengthened; colleges, universities, research institutes, and agricultural enterprises should be encouraged to conduct scientific research on other common problems and technical bottlenecks facing the current straw practices in the province to actively compensate for the shortcomings of existing technologies, explore more efficient straw field technology, and accelerate the speed of the application of scientific research results. Mature technologies in areas with conditions for pilot demonstrations should be selected to improve the agricultural situation in Heilongjiang, which is rich in straw resources, but the level of technology used in the field is low.

(6)  Establish a straw return service system

Universities and research institutes should take the lead in the use of the Internet and technology to establish a straw return service system, build a straw return platform, or establish intermediary service organizations that are market-oriented. Straw return enterprises and farmers should be contacted to provide corresponding services and solve the problem of the information asymmetry between farmers and enterprises.

(7)  Financial subsidies from the government

Currently, the common method of returning straw to the field involves the use of agricultural machinery to crush the straw and incorporate it into the field. The government can reasonably provide a certain number of financial subsidies according to the actual situation, as well as coordinate the corresponding departments, provide tax relief, and issue subsidies for agricultural machinery and other policies to promote the further development of environmentally friendly agriculture.

(8)  Return the full amount of straw to the field

According to the comprehensive benefit evaluation results, returning the full amount of straw to the field produces the highest economic, ecological, and social benefits. Returning the full amount of straw to the field can expand business opportunities in related industries, considerably improving the market economy and creating several jobs, avoiding too many rural people leaving the industry, and improving the labor shortages. Returning the full amount of straw to the field is important for alleviating the energy crisis and reducing environmental pollution.

## 6. Conclusions

Based on the characteristics of agricultural structure in different regions of Heilongjiang Province and the need to promote sustainable agricultural development and improve the agricultural ecological environment, in this study, we explored the comprehensive benefits of returning different volumes of crop straw to the field, constructed the related promotion system, and analyzed the selection of different crop straw return volumes in Heilongjiang. We analyzed the current situation and problems facing the practice of returning straw to fields for major crops, constructed an index system for evaluating the return of straw to fields, and evaluated the benefits of returning different amounts of straw to fields in Heilongjiang. The main study contents and conclusions are summarized as follows.

Theoretically, the existing assessment methods do not fully reflect the economic, ecological, and social benefits of returning different straw quantities to the field. This study further enriches the theoretical literature on straw return assessment indexes: we established comprehensive benefit assessment indices suitable for straw return quantities, guided by relevant theoretical studies. By using hierarchical analysis and the fuzzy comprehensive evaluation method, we constructed a comprehensive benefit assessment index considering three aspects, namely economic, ecological, and social benefits, including a total of ten items. We also specified the index weights to comprehensively assess the benefits of returning three different quantities of straw to the field in Heilongjiang Province. We found that the comprehensive benefit of returning the full amount of straw is the highest.

In practice, we found that problems arise in the selection of the amount of straw to return to the field in Heilongjiang through the statistical analysis of the major crop yields and the amount of major crop straw resources in the province in a ten-year period. Combined with the actual situation of returning crop straw to the field in Heilongjiang, we found that selecting the amount of straw to be returned to the field should be a focus in the future promotion of returning straw to the field. This method of comprehensively evaluating the benefit of returning various straw quantities to the soil in Heilongjiang can provide a scientific understanding of the various straw return quantities and their characteristics, which is conducive to the promotion and optimal adjustment of efficient straw return quantities. This method can be used to provide suggestions for relevant departments and farmers in the promotion and selection of straw return, thus promoting a scientific and rational straw return process in Heilongjiang Province, which is valuable for resource utilization and environmental protection.

**Author Contributions:** Conceptualization, J.M. and Z.Z.; methodology, X.L.; validation, H.Z.; formal analysis, C.L.; investigation, J.M. and Z.Z.; data curation, X.L.; writing—original draft preparation, J.M. and Z.Z.; writing—review and editing, J.M. and Z.Z.; visualization, C.L.; supervision, H.Z. All authors have read and agreed to the published version of the manuscript.

**Funding:** This research received no external funding.

**Institutional Review Board Statement:** Not applicable.

**Informed Consent Statement:** Not applicable.

**Data Availability Statement:** Not applicable.

**Conflicts of Interest:** The authors declare no conflict of interest.

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
