# Peer review of "Comprehensive Benefit of Crop Straw Return Volume under Sustainable Development Management Concept in Heilongjiang, China"

_sustainability, doi:10.3390/su15054129_

Round 1
Reviewer 1 Report
In the manuscript titled Evaluation on Comprehensive Benefit of Crop Straw Return Volume Under the Management Concept of Sustainable Development of Heilongjiang in China, Jia Mao, Ziang Zhao, Xiangyu Li, Honggang Zhao and Ciyun Lin estimated the number of straw resources in Heilongjiang province and used AHP-fuzzy integrated evaluation method to evaluate the comprehensive benefits of different straw return amounts. This study contains some interesting findings and are valuable for the understanding of sustainable development of the environment. This is a well-written paper containing interesting results which merit publication. For the benefit of the reader, however, a number of points need clarifying and certain statements require further justification. There are given below.
1. In the abstract and conclusion sections, it is recommended that the authors highlight the impact of straw return on sustainable development of the environment.
2. In the section of measuring straw resources in Chapter 3, the authors are invited to add curves or line graphs on the basis of the tables.
3. The language of the current manuscript needs to be polished.
4. The problem analysis in 3.5 is not detailed enough for each problem and needs to be added.
5. The font size in Figure 1 is too large and should be consistent with the font size of the full text.
6. Line 464-469, please add reference.

Author Response
Dear Reviewer:
Thanks for your letter and for the reviewers' comments concerning our manuscript. Those comments are valuable for revising and improving our paper with important guiding significance. We have made correction according to the comments, revisions are marked up using the “Track Changes” function. The responds to the reviewer's comments are as follows:
Comment 1: In the abstract and conclusion sections, it is recommended that the authors highlight the impact of straw return on sustainable development of the environment.
Response: In the abstract and conclusion sections, we have emphasized that straw return is beneficial to the sustainable development of the environment.
Comment 2: In the section of measuring straw resources in Chapter 3, the authors are invited to add curves or line graphs on the basis of the tables.
Response: For a better visual image, we have supplemented Figure 4 in Chapter 3 with a map of the cities.
Comment 3: The language of the current manuscript needs to be polished.
Response: We regret there were problems with the English. The paper has been carefully revised by a professional language editing service to improve the grammar and readability.
Comment 4: The problem analysis in 3.5 is not detailed enough for each problem and needs to be added.
Response: We have added the problem analysis section in 3.5.
Comment 5: The font size in Figure 1 is too large and should be consistent with the font size of the full text.
Response: We have adjusted the font size of Figure 1 to match the font size of the full text.
Comment 6: Line 464-469, please add reference.
Response: Thank you for your valuable suggestion. We have cited related literature in the proper place of the revised manuscript.

Reviewer 2 Report
This paper develops an AHP-fuzzy integrated evaluation method to evaluate the comprehensive benefits of different straw return amounts. It is an interesting research topic, and it can be published after following issues are resolved:
1. The research motivation and research gap should be addressed in the first section.
2. There are many evaluation research applications. The AHP-related evaluation methods and applications should be reviewed in Section 2.
3. What is the purpose of the section 3. There are two model sections in section 3 and section 4. It will be better if the links between these two sections are addressed.
4. It will be better if detail steps could be presented of the formulated model.
5. The English-editing needs to be improved and polished for better communication.
6. Theoretical contributions and practical implications should be highlighted in the discussion section.
7. There are vast majority of MCDM evaluation methods and applications, and a systematic review needs to be conducted, such as:
(1) Sustainable recycling partner selection using fuzzy DEMATEL-AEW-FVIKOR: A case study in small-and-medium enterprises (SMEs).
(2) Service quality evaluation of terminal express delivery based on an integrated SERVQUAL-AHP-TOPSIS approach.
(3)Prioritizing and overcoming biomass energy barriers: Application of AHP and G-TOPSIS approaches.
...
Author Response
Dear Reviewer:
Thanks for your letter and for the reviewers' comments concerning our manuscript. Those comments are valuable for revising and improving our paper with important guiding significance. We have made correction according to the comments, revisions are marked up using the “Track Changes” function. The responds to the reviewer's comments are as follows:
Comment 1: The research motivation and research gap should be addressed in the first section.
Response: Thank you for your valuable suggestion. We have added the research motivation and research gap in the second paragraph of the first section.
Comment 2: There are many evaluation research applications. The AHP-related evaluation methods and applications should be reviewed in Section 2.
Response: We have reviewed the AHP-related evaluation methods and applications in Section 2.
Comment 3: What is the purpose of the section 3. There are two model sections in section 3 and section 4. It will be better if the links between these two sections are addressed.
Response: The third section of this paper is an analysis of the current situation and problems of straw returning to fields in Heilongjiang Province. The purpose of Section 3 is to grasp the current situation of straw resources and straw returning to fields in Heilongjiang Province through data measurement, and finally find the current problems of straw returning to fields.
We have addressed the links between the two model sections. We have added a new section on suggestions for response in Section 5. This section have combined the current situation of straw return in Heilongjiang Province in Section 3 and the evaluation results of the amount of straw return in Section 4 and Section 5.
Comment 4: It will be better if detail steps could be presented of the formulated model.
Response: We gratefully appreciate for your valuable suggestion. We have given the detail steps of the formulated model. This paper that involves the formulated model is the construction of the evaluation index system and the specific steps of the evaluation method. First, we give the detailed steps of the evaluation index system construction in 4.1. Then, we give the specific steps of AHP to calculate the index weights in 4.2. Finally, we give the specific steps of the comprehensive benefit evaluation in 4.3. Based on these, we have made a specific evaluation in Section 5.
Comment 5: The English-editing needs to be improved and polished for better communication.
Response: We regret there were problems with the English. The paper has been carefully revised by a professional language editing service to improve the grammar and readability.
Comment 6: Theoretical contributions and practical implications should be highlighted in the discussion section.
Response: We have adjusted the conclusion section of the article. We have highlighted the theoretical contributions and practical implications of this paper at the end of the article.
Comment 7: There are vast majority of MCDM evaluation methods and applications, and a systematic review needs to be conducted, such as:
(1) Sustainable recycling partner selection using fuzzy DEMATEL-AEW-FVIKOR: A case study in small-and-medium enterprises (SMEs).
(2) Service quality evaluation of terminal express delivery based on an integrated SERVQUAL-AHP-TOPSIS approach.
(3) Prioritizing and overcoming biomass energy barriers: Application of AHP and G-TOPSIS approaches.
Response: We are so grateful for your kind suggestion. We have conducted a systematic review. We have carefully studied the articles you listed.
(1) This article developed a feasible analytical framework for recycling partner selection. It developed a novel integrated MCDM “fuzzy DEMATEL-AEW-FVIKOR” approach. We have supplemented the article with the fuzzy DEMATEL technique.
(2) This article developed an integrated SERVUQAL-AHP-TOPSIS approach. The evaluation index system including 19 criteria is established from the viewpoint of SERUQAL dimensions. The AHP technique is explored to determine the criteria weight of each indicator. The TOPSIS steps are used to prioritize the QoS level of alternative brands. We have supplemented the article with the AHP technique.
(3) This article used modified Delphi approach to prioritize biomass energy barriers. This article used the AHP approach to determine the weights of major barriers and sub-barriers. We have supplemented the article with the AHP approach.

Reviewer 3 Report
Dear authors,
the manuscript is presenting useful data in regard to circular economy needs for province of Heilongjiang in China.
The article is well documented, with research design clearly stated and results presented in a logical sequence.
However, the manuscript I suggest several improvements to be done, for a better quality of presentation.
Clearly state the research question and hypothesis in the introductory part.
English style needs improvement. Several sentences are too long (especially in introduction), with comma separated, making them difficult to follow.
Sections 2.1 and 2.2 have the same subtitle.
Methods section is now presented in various subsections. Merge them to a single section.
For a better visual image, table 6 needs to be supplemented with a map of the provinces.
Conclusions need to be re-arranged. Several paragraphs could be moved to discussions.
Best regards,
Author Response
Dear Reviewer:
Thanks for your letter and for the reviewers' comments concerning our manuscript. Those comments are valuable for revising and improving our paper with important guiding significance. We have made correction according to the comments, revisions are marked up using the “Track Changes” function. The responds to the reviewer's comments are as follows:
Comment 1: Clearly state the research question and hypothesis in the introductory part.
Response: We gratefully appreciate for your valuable suggestion. We have clearly stated the research question and hypothesis at the beginning of the second paragraph in the introductory part.
Comment 2: English style needs improvement. Several sentences are too long (especially in introduction), with comma separated, making them difficult to follow.
Response: We regret there were problems with the English. The paper has been carefully revised by a professional language editing service to improve the grammar and readability.
Comment 3: Sections 2.1 and 2.2 have the same subtitle.
Response: Thank you for your valuable suggestion. We have adjusted the subtitle of sections 2.1 and 2.2.
Comment 4: Methods section is now presented in various subsections. Merge them to a single section.
Response: We have merged methods sections in Section 2.
Comment 5: For a better visual image, table 6 needs to be supplemented with a map of the provinces.
Response: We have supplemented Figure 4 in Chapter 3 with a map of the cities from Table 6.
Comment 6: Conclusions need to be re-arranged. Several paragraphs could be moved to discussions.
Response: We have re-arranged the conclusions. We have moved the suggestions section of the conclusion to a separate section.

Reviewer 4 Report
The study is interesting because the suitable and sustainable land management is the need of today. The paper is written nicely but following suggestions could improve the paper.
- Introduction needs revision as suggested in the reviewed manuscript (attached herewith).
- please mention the knowledge gap in introduction.
- Conclusion need concreteness as it is too long.
- I have mentioned and marked some corrections in the paper for your concern, include these appropriately.

Author Response
Dear Reviewer:
Thanks for your letter and for the reviewers' comments concerning our manuscript. Those comments are valuable for revising and improving our paper with important guiding significance. We have made correction according to the comments, revisions are marked up using the “Track Changes” function. The responds to the reviewer's comments are as follows:
Comment 1: Introduction needs revision as suggested in the reviewed manuscript (attached herewith).
Response: Thank you so much for your careful check. We have revised the introduction as suggested in the reviewed manuscript.
Comment 2: Please mention the knowledge gap in introduction.
Response: We have mentioned the knowledge gap in introduction.
Comment 3: Conclusion need concreteness as it is too long.
Response: We have enhanced the concreteness of the conclusions. We have created a separate section for the suggestions section of the previous conclusions. We have reduced the content in the conclusion.
Comment 4: I have mentioned and marked some corrections in the paper for your concern, include these appropriately.
Response: We gratefully appreciate for your valuable suggestion. We have modified the corrections you marked. We have added content to the introduction and the corresponding references, and corrected the words you marked. We have modified the table title. We have concreted the conclusion. We have added the suggested references accordingly.

Round 2
Reviewer 2 Report
The paper is acceptble for its current version.